# RASP Quadratures: Efficient Numerical Integration for High-Dimensional Mean-Field Variational Inference

## Abstract

Efficient high-dimensional integration enables novel approaches to calibrate and control model uncertainty during training. In particular, the recently-proposed projective integral update formulation of variational inference derives model uncertainty from expectations that extract the local loss topography. Thus, we propose random-affinity sigma-point (RASP) quadratures, which are designed to eliminate integration errors from basis functions that drive Gaussian mean-field updates. Using only 3 gradient evaluations, RASP quadratures can extract locally-averaged gradients and Hessian diagonals from the loss, while eliminating errors from over half of all quadratic total-degree terms. Alternatively, we can use 6-point RASP quadratures to obtain $5^{\text{th}}$-order exactness in all univariate terms as well as $3^{\text{rd}}$-order exactness for two-thirds of bivariate terms. This work presents the design of RASP quadratures, theoretical guarantees on exactness, and analysis of expected errors. We also provide an open-source PyTorch implementation of RASP quadratures with quasi-Newton variational Bayes (QNVB), i.e. the projective integral update algorithm for Gaussian mean fields. Although RASP quadratures are designed to support QNVB, they are also compatible with other forms of variational inference, such as stochastic gradient variational Bayes (SGVB). Our experiments compare alternative integration schemes and training methods using three different learning tasks and architectures, demonstrating that efficient integration can improve generalizability for architectures with suitable loss structure.

## 1 Introduction

### 1.1 Motivation

Bayesian inference provides a foundational framework for reconciling new data with plausible predictive models and Markov-chain Monte Carlo (MCMC) (Metropolis et al., 1953; Hastings, 1970) is a well-known method to obtain posterior samples for prediction tasks. However, variational inference (VI) (Mézard et al., 1987; Parisi & Shankar, 1988) provides a needed alternative for large models and datasets, when the time and storage complexities of MCMC become unacceptable (Bishop & Nasrabadi, 2006; Jordan et al., 1999; Blei et al., 2017; Zhang et al., 2018).

Mean-field VI (MFVI) (Anderson & Peterson, 1987; Peterson & Hartman, 1989; Hinton & Van Camp, 1993) provides an alternative means to capture model uncertainty in large learning models by approximating a local component of the full posterior. Mean-field distributions represent uncertainty as a product of independent distributions in each parameter, reducing storage to a small constant times the number of parameters. Other forms of VI are possible (Dhaka et al., 2021), Gaussian mean-fields provide a simple method to calibrate local parameter uncertainty and improve predictions by taking small model variations into account. Localized uncertainty also supports adaptive rounding schemes to reduce representational complexity after training (Banner et al., 2019; Choukroun et al., 2019; Cai et al., 2020; Nagel et al., 2020).

Projective integral updates (Duersch, 2023) provide an efficient framework for updating variational densities using local integrals to project the posterior onto a corresponding basis of functions. For Gaussian mean-fields, these updates yield the quasi-Newton variational Bayes (QNVB) algorithm,

which analytically recovers quasi-Newton steps as locally optimal updates. However, these updates require the potentially noisy or expensive computation of the expected gradient and Hessian diagonal of the training loss. To enhance the projective integral update framework, particularly in resource-constrained settings when each gradient is expensive, we desire more efficient methods to reduce evaluations and minimizing projection errors for components that dominate the VI update.

## 1.2 CONTRIBUTIONS

This work proposes random-affinity sigma-point (RASP) quadratures, an efficient integration approach for mean-field densities that supports projective integral updates by minimizing the error corresponding to basis functions that dominate variational updates. This framework allows us to approximate the locally-averaged gradient and Hessian using ordinary backpropagation to support uncertainty calibration during training. RASP quadratures are also compatible with other VI algorithms, such as stochastic gradient variational Bayes (Kingma & Welling, 2013).

RASP quadratures can eliminate integration errors from all univariate quadratics and half of all bivariate quadratic basis functions using only 3 function evaluations. Similarly, we can achieve fifth-order exactness in univariate polynomials and two-thirds of bivariate quadratic basis functions using 6 function evaluations. These results are further substantiated by our experiments, which show that different learning architectures benefit from QNVB with RASP quadratures to achieve competitive test accuracy against multiple baselines. Efficient methods for high-dimensional integration facilitate training with model uncertainty while using very few function evaluations, making them ideal for resource-constrained environments.

The main contributions of this work (Section 3) include:

1. the design and error analysis of RASP quadratures (Section 3),
2. comparisons to recent work (Section 4), and
3. an open-source PyTorch implementation of QNVB with RASP quadratures[1].

## 2 BACKGROUND

### 2.1 VARIATIONAL INFERENCE

Here, we briefly review VI. (Jordan et al., 1999; Blei et al., 2017; Zhang et al., 2018) provide more thorough overviews. VI addresses the intractability of Bayesian inference in high-dimensional model classes. A dataset $\mathcal{D}$ comprising ordered pairs of features $\boldsymbol{x}$ and labels $\boldsymbol{y}$ informs a distribution over plausible models. Each model $\boldsymbol{\theta} \in \mathbb{R}^d$ acts on inputs to generate a distribution over label predictions $\mathbf{p}(\boldsymbol{y} \mid \boldsymbol{\theta}, \boldsymbol{x})$. By taking the product over all data samples, we obtain the likelihood $\mathbf{p}(\mathcal{D} \mid \boldsymbol{\theta})$. Given a prior over models $\mathbf{p}(\boldsymbol{\theta})$, Bayes' theorem yields the posterior $\mathbf{p}(\boldsymbol{\theta} \mid \mathcal{D}) \propto \mathbf{p}(\mathcal{D} \mid \boldsymbol{\theta})\mathbf{p}(\boldsymbol{\theta})$.

Unfortunately, parameter regions that dominate the posterior are difficult to locate and store when the number of trainable parameters $d$ is high. Thus, we only seek to capture a component of the posterior using a simpler variational distribution $\mathbf{q}(\boldsymbol{\theta} \mid \boldsymbol{\varphi})$, where the variational parameters $\boldsymbol{\varphi}$ describe its shape.

Mean-field densities are products of independent factor densities in each parameter, $\mathbf{q}(\boldsymbol{\theta} \mid \boldsymbol{\varphi}) = \prod_{i=1}^{d} \mathbf{q}(\boldsymbol{\theta}_i \mid \boldsymbol{\varphi}_i)$, where each $\boldsymbol{\varphi}_i$ is a block of shape parameters that describe uncertainty in $\boldsymbol{\theta}_i$. This structure makes mean fields an attractive choice for the family of variational densities in high dimensions, because they support efficient optimization, storage, and integration. These properties ultimately serve to improve the feasibility of obtaining more robust predictions from the variational-predictive integral, which is defined as

$$\mathbf{q}(\boldsymbol{y} \mid \boldsymbol{x}, \boldsymbol{\varphi}) = \int \mathbf{p}(\boldsymbol{y} \mid \boldsymbol{x}, \boldsymbol{\theta}) \, d\mathbf{q}(\boldsymbol{\theta} \mid \boldsymbol{\varphi}). \tag{1}$$

Optimization minimizes the KL-divergence from the posterior to the variational distribution:

$$\boldsymbol{\varphi}^* = \underset{\boldsymbol{\varphi}}{\arg\min} \int d\mathbf{q}(\boldsymbol{\theta} \mid \boldsymbol{\varphi}) \log\left(\frac{\mathbf{q}(\boldsymbol{\theta} \mid \boldsymbol{\varphi})}{\mathbf{p}(\boldsymbol{\theta} \mid \mathcal{D})}\right). \tag{2}$$

[1][anonymized URL]

## 2.2 PROJECTIVE INTEGRAL UPDATES

Projective integral updates (PIU) (Duersch, 2023) optimize the variational density by computing expectations over a compatible basis of functions $\mathcal{F}$, provided $\mathrm{span}(\mathcal{F})$ contains every log-density in the variational family, i.e.

$$\log \mathbf{q}(\boldsymbol{\theta} \mid \boldsymbol{\varphi}) = \sum_{\ell=0}^{m} \varphi_\ell f_\ell(\boldsymbol{\theta}) \quad \text{where} \quad \mathcal{F} = \{ f_0(\boldsymbol{\theta}) = 1 \text{ and } f_\ell(\boldsymbol{\theta}) \text{ for } \ell \in [m] \}. \tag{3}$$

By using the variational density at training step $t$, i.e. $\mathbf{q}(\boldsymbol{\theta} \mid \boldsymbol{\varphi}^{(t)})$, to define an inner product on functions, we can obtain each projection coefficient $\varphi_\ell^{(t+1)}$ for $\ell \in [m]$ with the fixed-point iteration

$$\varphi_\ell^{(t+1)} = \frac{\langle f_\ell, \log \mathbf{p}(\mathcal{D} \mid \boldsymbol{\theta})\mathbf{p}(\boldsymbol{\theta}) \rangle_{\boldsymbol{\varphi}^{(t)}}}{\langle f_\ell, f_\ell \rangle_{\boldsymbol{\varphi}^{(t)}}} \quad \text{where} \quad \langle f, g \rangle_{\boldsymbol{\varphi}} \equiv \int f(\boldsymbol{\theta}) g(\boldsymbol{\theta}) \, d\mathbf{q}(\boldsymbol{\theta} \mid \boldsymbol{\varphi}) \tag{4}$$

and normalization determines $\varphi_0$. The optimum of Equation (2) is a fixed point of Equation (4).

When projective integral updates are applied to Gaussian mean fields, denoted as $\mathbf{q}(\boldsymbol{\theta} \mid \boldsymbol{\mu}, \boldsymbol{\sigma}^2) \equiv \mathcal{N}(\boldsymbol{\theta} \mid \boldsymbol{\mu}, \mathrm{diag}(\boldsymbol{\sigma})^2)$, the resulting algorithm is called quasi-Newton variational Bayes (QNVB). In this case, a suitable orthogonal basis $\mathcal{F}$ contains $m = 2d + 1$ functions, taking the form

$$\mathcal{F} = \left\{ f_0(\boldsymbol{\theta}) = 1, \ f_{2i-1}(\boldsymbol{\theta}) = \boldsymbol{\theta}_i - \boldsymbol{\mu}_i, \ f_{2i}(\boldsymbol{\theta}) = (\boldsymbol{\theta}_i - \boldsymbol{\mu}_i)^2 - \boldsymbol{\sigma}_i^2 \mid i \in [d] \right\}. \tag{5}$$

In this case, the projection coefficients are equivalent to the expected gradient $\boldsymbol{g}$ and Hessian diagonal $\boldsymbol{h}$ of the loss (i.e., the unnormalized negative log posterior), which we can write as

$$\mathcal{L}(\boldsymbol{\theta} \mid \mathcal{D}) \equiv -\log \mathbf{p}(\mathcal{D} \mid \boldsymbol{\theta})\mathbf{p}(\boldsymbol{\theta}) \approx \mathcal{L}_\mu + (\boldsymbol{\theta} - \boldsymbol{\mu})^T \left[ \boldsymbol{g} + \frac{1}{2} \boldsymbol{h} * (\boldsymbol{\theta} - \boldsymbol{\mu}) \right]. \tag{6}$$

The constant $\mathcal{L}_\mu$ approximates the loss at $\boldsymbol{\mu}$, the gradient $\boldsymbol{g} \in \mathbb{R}^d$ captures linear perturbations, and the Hessian diagonal $\boldsymbol{h} \in \mathbb{R}^d$ captures quadratic effects. Note that $*$ is the Hadamard (elementwise) product and powers are also elementwise, e.g. $\boldsymbol{h}^{-1} * \boldsymbol{g} = \mathrm{diag}(\boldsymbol{h})^{-1}\boldsymbol{g}$. This local quadratic approximation of the loss topography updates the variational density with a quasi-Newton step, i.e. $\boldsymbol{\mu} \leftarrow \boldsymbol{\mu} - \boldsymbol{h}^{-1} * \boldsymbol{g}$ and $\boldsymbol{\sigma} \leftarrow \boldsymbol{h}^{-1/2}$. Since both $\boldsymbol{g}$ and $\boldsymbol{h}$ can be computed from simple integrals:

$$\boldsymbol{g} = \int \nabla_{\boldsymbol{\theta}} \mathcal{L}(\boldsymbol{\theta} \mid \mathcal{D}) \, d\mathbf{q}(\boldsymbol{\theta} \mid \boldsymbol{\mu}, \boldsymbol{\sigma}^2) \quad \text{and} \tag{7}$$

$$\boldsymbol{h} = \boldsymbol{\sigma}^{-2} * \int (\boldsymbol{\theta} - \boldsymbol{\mu}) * \nabla_{\boldsymbol{\theta}} \mathcal{L}(\boldsymbol{\theta} \mid \mathcal{D}) \, d\mathbf{q}(\boldsymbol{\theta} \mid \boldsymbol{\mu}, \boldsymbol{\sigma}^2), \tag{8}$$

efficient integration for high-dimensional mean-fields may enhance PIU optimization.

## 2.3 QUADRATURES AND SIGMA POINTS

Numerical quadratures approximate weighted integrals by taking linear combinations of specific function evaluations. In this work, we consider functions $f : \mathbb{R}^d \to \mathbb{R}$ over the model parameter domain and seek to approximate integrals against $\mathbf{q}(\boldsymbol{\theta} \mid \boldsymbol{\mu}, \boldsymbol{\sigma}^2)$. In this setting, quadrature formulas take the form

$$Q[f] = \sum_{k=1}^{n} w_k f\left(\boldsymbol{\theta}^{(k)}\right) \approx \int f(\boldsymbol{\theta}) \, d\mathbf{q}(\boldsymbol{\theta} \mid \boldsymbol{\mu}, \boldsymbol{\sigma}^2). \tag{9}$$

Each evaluation index $k \in [n]$ has a weight $w_k \in \mathbb{R}$ and an evaluation node $\boldsymbol{\theta}^{(k)} \in \mathbb{R}^d$. These weights and nodes are solved to exactly integrate some basis $\mathcal{F}_\varepsilon$, so that

$$Q[f_\ell] = \int f_\ell(\boldsymbol{\theta}) \, d\mathbf{q}(\boldsymbol{\theta} \mid \boldsymbol{\mu}, \boldsymbol{\sigma}^2) \quad \text{for all} \quad f_\ell \in \mathcal{F}_\varepsilon. \tag{10}$$

Since both the weighted integral and the quadrature are linear functionals, it follows that the quadrature is exact for all $f \in \mathrm{span}(\mathcal{F}_\varepsilon)$. Thus, it is beneficial to construct a quadrature for which $\mathcal{F} \subset \mathcal{F}_\varepsilon$ from Equation (5). This will suppress errors in $\boldsymbol{g}$ and $\boldsymbol{h}$ by collecting a set of gradient evaluations from standard backpropagation, i.e.

$$\boldsymbol{g} \approx Q[\nabla_{\boldsymbol{\theta}} \mathcal{L}(\boldsymbol{\theta} \mid \mathcal{D})] \quad \text{and} \quad \boldsymbol{h} \approx \boldsymbol{\sigma}^{-2} * Q[(\boldsymbol{\theta} - \boldsymbol{\mu}) * \nabla_{\boldsymbol{\theta}} \mathcal{L}(\boldsymbol{\theta} \mid \mathcal{D})]. \tag{11}$$

For quadrature formulas that obtain second-order exactness over multivariate Gaussians, the evaluation nodes are called sigma points in the non-linear filtering literature. These formulas are used to propagate uncertainty in unscented Kalman filters (Uhlmann, 1995). The symmetric set used by Uhlmann, see (McNamee & Stenger, 1967), comprises $2d + 1$ nodes. Unfortunately, even minimal sigma-point quadratures require $d + 1$ nodes (Wan-Chun et al., 2007), which is infeasible for large $d$.

## 3 RANDOM-AFFINITY SIGMA-POINT (RASP) QUADRATURES

### 3.1 COMPOSITION

RASP quadratures $Q[\cdot]$ are generated by first forming a reference quadrature $R[\cdot]$ that acts on functions in a reference domain with $r$ dimensions. The exactness of $R[\cdot]$ impacts that of $Q[\cdot]$ and Section 3.2 provides a detailed analysis of this relationship.

Using weights $w_k$ and nodes $\boldsymbol{\rho}^{(k)} \in \mathbb{R}^r$ for each evaluation $k \in [n]$, we have the reference quadrature

$$R[f] = \sum_{k=1}^{n} w_k f(\boldsymbol{\rho}^{(k)}) \approx \int f(\boldsymbol{\rho}) \, d\mathcal{N}(\boldsymbol{\rho} \mid 0, \boldsymbol{I}). \tag{12}$$

In principle, $R[\cdot]$ could be any quadrature formula (also called cubature in multiple dimensions), but the number of evaluations $n$ will determine the number of gradients computed from backpropagation per training step. Thus, sigma points are preferred because they achieve second-order exactness with few evaluations, but higher-order quadratures exist (McNamee & Stenger, 1967; Smolyak, 1963; Novak & Ritter, 1999; Dick et al., 2013; Menegaz et al., 2015). Algorithms 1 and 2 (see Appendix A.1) generate reference quadratures from the simplex- and cross-polytope vertices, respectively.

We can then generate a RASP quadrature $Q[\cdot]$ by randomly mapping each parameter dimension to a signed reference dimension. For each $i \in [d]$, select a reference dimension $j_i \in [r]$ uniformly at random (with replacement) along with a random sign $s_i \in \{-1, 1\}$. These random parameter affinities with reference dimensions yield the RASP quadrature

$$Q[f] = \sum_{k=1}^{n} w_k f(\boldsymbol{\mu} + \boldsymbol{\sigma} * \boldsymbol{x}^{(k)}) \approx \int f(\boldsymbol{\theta}) \, d\mathbf{q}(\boldsymbol{\theta} \mid \boldsymbol{\mu}, \boldsymbol{\sigma}^2) \quad \text{where} \quad \boldsymbol{x}_i^{(k)} = s_i \boldsymbol{\rho}_{j_i}^{(k)}. \tag{13}$$

**Memory management**  A memory-efficient implementation tracks random signs by doubling the number of reference dimensions. Each reference node is concatenated with its negative, i.e. $\widehat{\boldsymbol{\rho}}^{(k)T} = \left[ \boldsymbol{\rho}^{(k)T} \; -\boldsymbol{\rho}^{(k)T} \right]$, so that both the sign and reference index can be constructed from a single array of integers in $[2r]^d$. Algorithm 3 shows how this technique is implemented. For large architectures, this approach avoids excessive memory by generating each evaluation node only when it is needed. In contrast, the quasi Monte Carlo methods tested in Section 4 match sample moments to the distribution (Caflisch, 1998), which requires sampling, storing, and modifying $n \times d$ floating-point numbers simultaneously, rather than one node at a time.

### 3.2 EXACTNESS

To analyze the exactness properties of $Q[\cdot]$, we distinguish degrees of exactness $\varepsilon_1$ and $\varepsilon_2$ for univariate and bivariate polynomials, respectively. That is, $\varepsilon_1$ is the maximum polynomial degree for functions that depend only on a single coordinate, $f(\boldsymbol{\rho}) \equiv p(\boldsymbol{\rho}_j)$, that satisfies

$$R[f] = \int f(\boldsymbol{\rho}) \, d\mathcal{N}(\boldsymbol{\rho} \mid 0, \boldsymbol{I}). \tag{14}$$

Likewise, for functions that are polynomial in two coordinates, i.e $f(\boldsymbol{\rho}) \equiv p(\boldsymbol{\rho}_{j_1}, \boldsymbol{\rho}_{j_2})$, let $\varepsilon_2$ be the maximum total degree of $p$ for which Equation (14) holds.

For example, Algorithm 1 uses $n = r + 1$ nodes to obtain degrees of exactness $\varepsilon_1 = \varepsilon_2 = 2$. Alternatively, Algorithm 2 generates $n = 2r$ symmetric nodes. Here, symmetry means that for each $k \in [n]$, there exists $k' \in [n]$ for which $\boldsymbol{\rho}^{(k')} = -\boldsymbol{\rho}^{(k)}$ and $w_{k'} = w_k$. This property increases the degree of exactness to $\varepsilon_1 = \varepsilon_2 = 3$. It is also worth noting that when $r = 3$, we obtain univariate exactness $\varepsilon_1 = 5$.

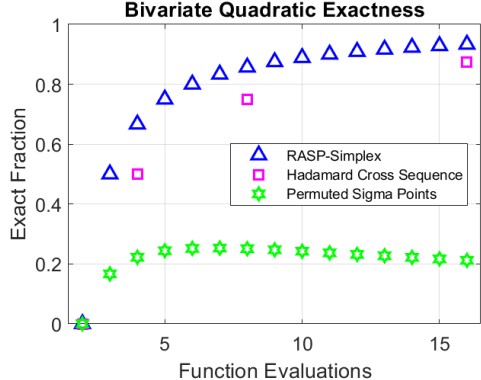

Figure 1: Fraction of quadratic basis functions that integrate exactly with different methods. RASP-simplex quadratures attain the largest exact fraction for a fixed number of nodes.

Theorems 3.1 and 3.2 explain how the exactness of $R[\cdot]$ influences the exactness of $Q[\cdot]$. Figure 1 shows the fraction of quadratic basis functions that are exactly integrated for different quadrature methods, including some proposed by Duersch (2023).

Although Theorem 3.2 implies that a fraction of bivariate terms will not be exact, Theorem 3.3 shows that even in such cases, the expected error is zero. Further, when the reference quadrature is symmetric, Theorem 3.4 shows that *every* odd total-degree term (centered at $\boldsymbol{\mu}$) becomes exact.

**Theorem 3.1** *Let $R[\cdot]$ be a reference quadrature, as in Equation* (12)*, with univariate exactness $\varepsilon_1$. Every RASP quadrature $Q[\cdot]$, from Equation* (13)*, also has univariate exactness $\varepsilon_1$.*

**Theorem 3.2** *Let $R[\cdot]$ be a reference quadrature with bivariate exactness $\varepsilon_2$. The probability that a RASP quadrature $Q[\cdot]$ has the same bivariate exactness in two parameters $\boldsymbol{\theta}_{i_1}$ and $\boldsymbol{\theta}_{i_2}$ is $\frac{r-1}{r}$.*

**Theorem 3.3** *For bivariate quadratics of the form $f(\boldsymbol{\theta}) = (\boldsymbol{\theta}_{i_1} - \boldsymbol{\mu}_{i_1})(\boldsymbol{\theta}_{i_2} - \boldsymbol{\mu}_{i_2})$, the expected error of $Q[f]$ is zero.*

**Theorem 3.4** *If $R[\cdot]$ is a symmetric reference quadrature then every RASP quadrature $Q[\cdot]$ exactly integrates all odd total-degree basis functions of the form*

$$f(\boldsymbol{\theta}) = \prod_{i=1}^{d}(\boldsymbol{\theta}_i - \boldsymbol{\mu}_i)^{m_i} \quad where \quad \sum_{i=1}^{d} m_i \quad is\ odd. \tag{15}$$

### 3.3 ERROR COMPARISONS

To gain insight into the errors associated with different integration methods, we examine a third-order expansion of the loss topography (simplified by setting $\boldsymbol{\mu} = 0$) and the corresponding gradient, i.e.

$$\mathcal{L}(\boldsymbol{\theta}) \approx \mathcal{L}_\mu + \sum_{i=1}^{d} \boldsymbol{\theta}_i \left( \boldsymbol{g}_i + \sum_{j=1}^{d} \frac{\boldsymbol{\theta}_j}{2} \left[ \boldsymbol{H}_{ij} + \sum_{k=1}^{d} \frac{\boldsymbol{\theta}_k}{3} \mathcal{T}_{ijk} \right] \right),$$

$$\text{which gives} \quad \partial_i \mathcal{L}(\boldsymbol{\theta}) \approx \boldsymbol{g}_i + \sum_{j=1}^{d} \boldsymbol{\theta}_j \left[ \boldsymbol{H}_{ij} + \sum_{k=1}^{d} \frac{\boldsymbol{\theta}_k}{2} \mathcal{T}_{ijk} \right]. \tag{16}$$

This expansion gives the exact expected gradient and Hessian diagonal in element $i$,

$$\mathbb{E}_{\mathbf{q}(\boldsymbol{\theta}|0,\,\boldsymbol{\sigma}^2)} \left[\partial_i \mathcal{L}(\boldsymbol{\theta})\right] = \boldsymbol{g}_i + \sum_{j=1,\,j\neq i}^{d} \mathcal{T}_{ijj} \boldsymbol{\sigma}_j^2 \quad \text{and} \quad \mathbb{E}_{\mathbf{q}(\boldsymbol{\theta}|0,\,\boldsymbol{\sigma}^2)} \left[ \frac{\boldsymbol{\theta}_i \partial_i \mathcal{L}(\boldsymbol{\theta})}{\boldsymbol{\sigma}_i^2} \right] = \boldsymbol{H}_{ii}. \tag{17}$$

Although RASP quadratures recover univariate contributions, e.g. $Q[\mathcal{T}_{ijj}\boldsymbol{\theta}_j^2] = \mathcal{T}_{ijj}\boldsymbol{\sigma}_j^2$, they also encounter errors from some cross terms. For gradient and Hessian expectations, these errors are

$$Q[\mathcal{T}_{ijk}\boldsymbol{\theta}_j\boldsymbol{\theta}_k] = \pm\mathcal{T}_{ijk}\boldsymbol{\sigma}_j\boldsymbol{\sigma}_k \quad \text{and} \quad Q\left[H_{ij}\frac{\boldsymbol{\theta}_i\boldsymbol{\theta}_j}{\boldsymbol{\sigma}_i^2}\right] = \pm H_{ij}\frac{\boldsymbol{\sigma}_j}{\boldsymbol{\sigma}_i} \quad \text{for some} \quad i,\,j,\,k \in [d]. \tag{18}$$

By accounting for the probability that bivariate quadratic terms are not exact, we obtain the variances shown in Appendix A.2. Since $n \geq r+1$, we see that Monte Carlo integration yields lower variance for cross terms. Thus, different choices of integration strategy may be superior for different architectures, depending on the relative weight of univariate terms to bivariate terms. See Appendix A.2 in Appendix A.2 for a table of error comparisons.

RASP quadratures aim to reduce errors on basis function integrals that dominate QNVB updates, but also incur persistent errors on higher-order basis functions. While the resulting estimator is not unbiased, our experiments show that RASP quadratures produce competitive results.

## 4 EXPERIMENTS

These experiments compare RASP quadratures with Monte Carlo (MC) and Variance-Reduced Monte Carlo (VRMC) integration for two VI optimization algorithms: stochastic gradient variational Bayes (SGVB) (Kingma & Welling, 2013; Ranganath et al., 2014; Titsias & Lázaro-Gredilla, 2014) and QNVB (Duersch, 2023). VRMC-1 translates Monte Carlo samples to match the sample mean to $\boldsymbol{\mu}$ (first-order exactness). Likewise, VRMC-2 also scales the samples to match the variance $\mathrm{diag}(\boldsymbol{\sigma})^2$ (univariate second-order exactness). See related work in Section 5.

We measure performance across a number of learning tasks and architectures: 1. ResNet18 (He et al., 2016) with CIFAR-10 (Krizhevsky & Hinton, 2009) for image classification, 2. Deep Learning Recommendation Model (DLRM) (Naumov et al., 2019) with the Criteo Ad Kaggle dataset for ad recommendation, and 3. Tensorized Transformer (Ma et al., 2019) with the Penn Treebank (PTB) (Marcus et al., 1993) dataset for text completion.

While our experiments use flat priors, many other priors would be equivalent to regularization terms in the negative log posterior. Notably, Fortuin et al. (2021) shows that wide-tailed distributions (e.g., $L_1$ regularization) are usual more effective for neural networks than Gaussian priors (i.e., $L_2$). Both SGVB and QNVB are compatible with regularization by simply adding the desired terms to the loss. In the context of Bayesian inference, a cross-validation search for optimal hyperparameters can be understood as a hierarchical model for which the hyperparameters are chosen by finding the maximum likelihood estimator.

MC, VRMC, and RASP quadratures are tested with 3, 4, and 6 evaluations each, excluding cases where the quadrature does not exist for the given number of evaluations. Each trial begins by setting a random seed for reproducibility. Therefore, the only source of variation across each set of experiments comes from the training method. For each set of seeds, we present the optimal validation run (bolded line) and full range of outcomes (shaded area). See Appendix A.3 for prediction quality comparisons with other ML training algorithms. These experiments show that the quadratures we compare have a more significant impact on QNVB than SGVB.

**ResNet18 / CIFAR-10**  Figure 2 and Figure 3 show that the choice of integration method has a marked effect on the optimization trajectory. SGVB does not appear to benefit from RASP quadratures, but the range of outcomes is worse than those for QNVB. For QNVB, the range of accuracy outcomes is significantly higher for all first-moment matched methods (VRMC and RASP) than MC. Notably, the RASP-simplex outperforms VRMC-1 and uses less memory than both VRMC-1 and VRMC-2.

**DLRM / Criteo Ad Kaggle**  The DLRM architecture we tested contains approximately 540 million parameters, causing VRMC methods exceed GPU memory limits. Thus, we only compare Monte Carlo, RASP-simplex, and RASP-cross quadratures. Figures 4 and 5 shows that the Monte Carlo quadrature outperforms the RASP quadratures for this architecture. This may be due to the weight of bivariate error terms in the gradient, which exhibit larger variance with RASP quadratures than MC (see Appendix A.2).

**Tensorized Transformer / PTB**  The 3-evaluation quadrature comparisons in Figure 7 show that the RASP-simplex reduces the validation loss much earlier, and to a lower minimum, than alternatives. The other quadratures can achieve comparable validation loss, but require more gradient evaluations. We also compare QNVB (using RASP-simplex 3) to other training algorithms in Figure 10 and find that it achieves better validation perplexity. In particular, it achieves a significantly lower perplexity of $9.45$, whereas the next lowest perplexity is $45.67$ with SGVB.

## 5 RELATED WORK

A large body of work focuses on reducing integration errors using Quasi Monte Carlo (QMC) methods. For overviews, see Morokoff & Caflisch (1994); Caflisch (1998); Dick et al. (2013); Leobacher &

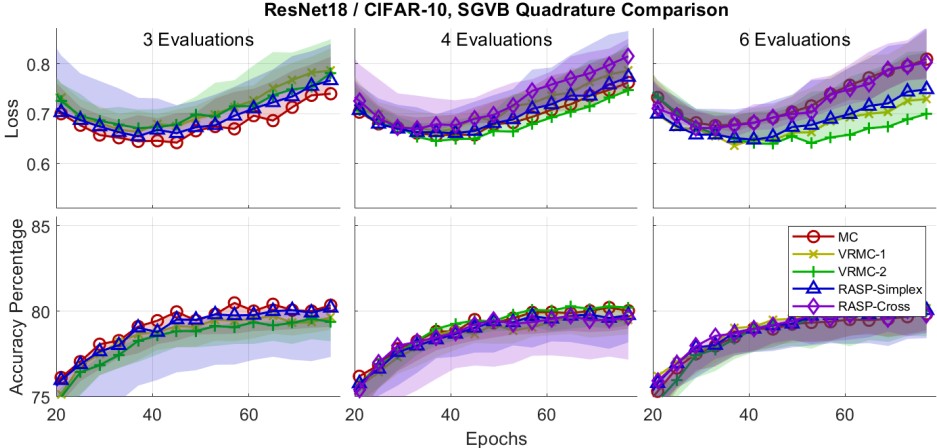

Figure 2: We compare quadratures on ResNet18 trained with SGVB for 3, 4, and 6 evaluations. RASP quadratures do not appear to offer any benefit for this architecture and training algorithm. Validation optima are achieved with MC and VRMC-2.

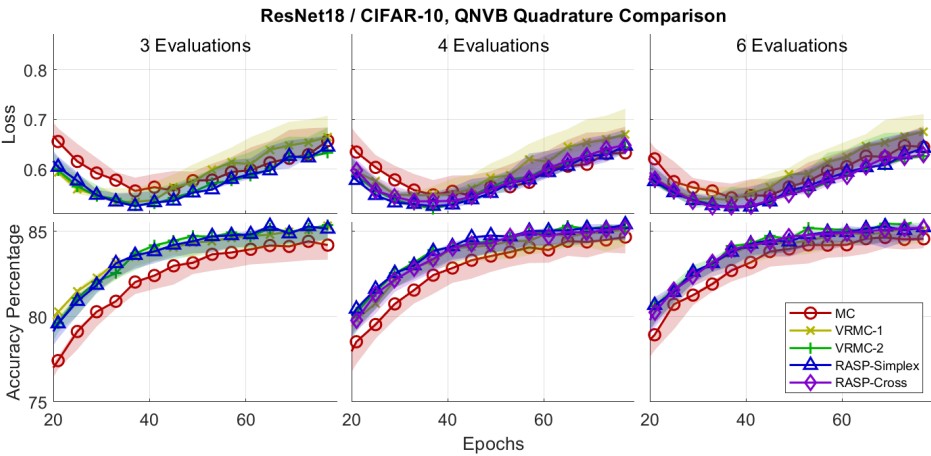

Figure 3: We compare quadratures on ResNet18 trained with QNVB for 3, 4, and 6 evaluations. QNVB offers a clear improvement in the range of outcomes. There is clear separation between Monte Carlo and first-moment matched methods. RASP quadratures frequently achieve peak performance and use less memory than VRMC.

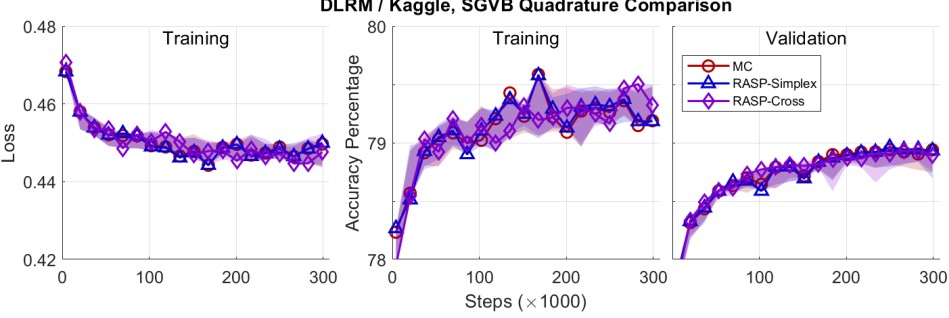

Figure 4: We compare computationally-feasible quadratures on the DLRM architecture trained with SGVB. Technically, the validation-optimum is achieved by RASP simplex 3, but only very slightly. RASP quadratures do not appear to offer any notable benefit for this training algorithm.

Pillichshammer (2014). QMC improves expectation approximations by generating samples from the unit hypercube $[0, 1]^d$ that are roughly evenly distributed. Randomized variants (RQMC) further protect against the worst-case errors that QMC can encounter (L'Ecuyer, 2018; Dick et al., 2022;

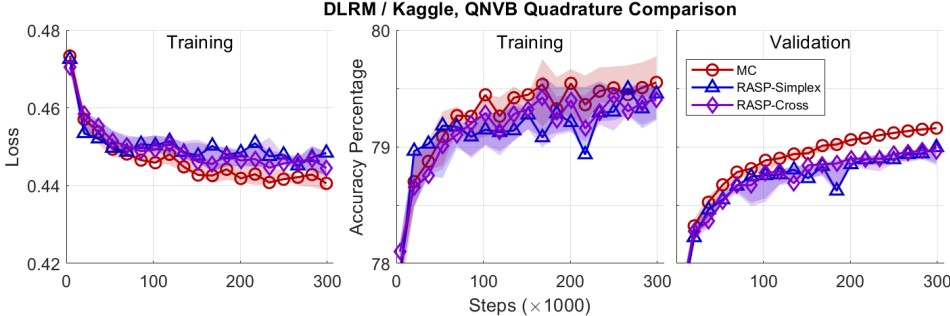

Figure 5: Here we compare the same quadratures on the DLRM architecture trained with QNVB. The loss characteristics of DLRM drive the Monte Carlo quadratures to achieve better performance.

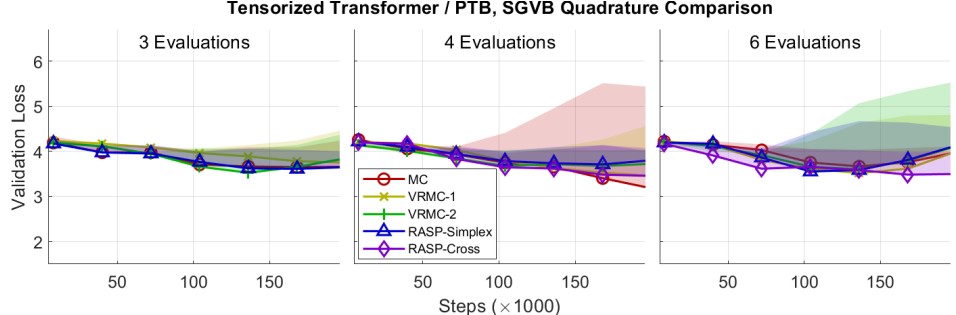

Figure 6: We compare quadratures using 3, 4, and 6 evaluations on the Tensorized Transformer with SGVB. RASP quadratures offer no significant benefit for SGVB.

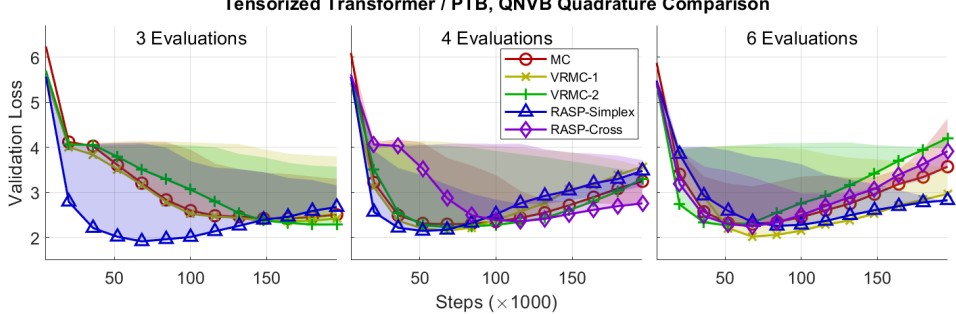

Figure 7: We compare the same quadratures on the Tensorized Transformer trained with QNVB. RASP-Simplex 3 reduces the loss early on and achieves a lower minimum at 3 evaluations.

Owen, 2023). Efforts to incorporate RQMC into VI (Buchholz et al., 2018; Liu & Owen, 2021) seek to improve training by improving the convergence of general integrals and Lin et al. (2022) provide explicit convergence guarantees.

The success of (R)QMC methods for many high-dimensional integration problems is somewhat unexpected. As Trefethen (2017) explains, integrals that are dominated by terms with complexity that depends on only a few important directions should perform well, but rotating such functions produces problems due to the anisotropy of the hypercube. In contrast, this work aims to minimize function evaluations by incorporating low-dimensional symmetry into high-dimensional integration. Our experiments demonstrate competitive results with only 3 RASP-simplex evaluations. In comparison, tests by Buchholz et al. (2018) use 10 samples and Liu & Owen (2021) performs convergence experiments from 8 to 2048 samples.

Our approach leverages the fact the integrals comprising projective integral updates (Duersch, 2023) are dominated by specific basis functions. Rather than enhancing convergence of arbitrary integrals, we only seek to reduce errors associated with the basis functions needed for QNVB updates. Since VRMC methods (Caflisch, 1998) also yield $1^{st}$-order and univariate $2^{nd}$-order exactness, they provide

a useful comparison against RASP quadratures for Gaussian MFVI. Nevertheless, it may be beneficial to examine modified RQMC methods that target exactness on second-order basis functions, while also retaining guarantees on convergence.

To the best of our knowledge, the RASP method of generating high-dimensional quadratures from sigma points has not been previously proposed. Sigma points take advantage of isotropy and are specifically designed for Gaussian integration. Menegaz et al. (2015) surveys sigma-point methods, which have not focused on applications where $n = d + 1$, the minimal sigma-point rule (Wan-Chun et al., 2007), is prohibitive. Since then, Radhakrishnan et al. (2018) have proposed a new sigma-point method using $4r + 1$ evaluations that reduces high-order sensitivity by increasing the evaluation weights near the center of the density. The recent book by Dick et al. (2022) also covers high-dimensional numerical integration with lattices and QMC methods on the unit hypercube. If a method similar to ours has been proposed before, it is not commonly discussed.

## 6 CONCLUSION

RASP quadratures provide an evaluation-efficient method for high-dimensional Gaussian mean-field integration that can improve model uncertainty calibration. Specifically, RASP quadratures minimize the error corresponding to basis functions that dominate variational updates for Gaussian mean fields. We presented the design of RASP quadratures and memory-efficient implementations, with a link to our PyTorch implementation. We also proved theoretical guarantees of exactness and expected errors, showing how RASP quadratures are more efficient than Monte Carlo for univariate basis functions. Our results demonstrate that RASP quadratures can eliminate integration error from all univariate quadratics and half of all bivariate quadratic basis functions using only 3 evaluations. Further, we experimentally demonstrated the efficacy of RASP quadratures. In tests on ResNet18 and Tensorized Transformer architectures, RASP quadratures frequently achieved peak performance while using less memory than VRMC methods. QNVB with select RASP quadratures achieved higher accuracy and lower perplexity than several state-of-the-art training algorithms.

RASP quadratures offer a reliable numerical approximation of the local loss topography at a manageable cost, enabling superior error control with only a few model evaluations and standard backpropagation. RASP quadratures may support more advanced training methodologies by informing the effect that potential parameter perturbations have on prediction quality. Similarly, computationally-limited learning tasks and *in situ* data processing, including power-limited control systems, may benefit from reducing the number of model evaluations needed to propagate uncertainty.

ACKNOWLEDGMENTS

[anonymized acknowledgements]

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

# A  APPENDIX

## A.1  ALGORITHMS

---

**Algorithm 1** Simplex vertices, invariant under coordinate permutations

---

**Input:** $r \in \mathbb{Z}_{\geq 1}$ is the number of reference dimensions.
**Output:** $\boldsymbol{R}$ is an $r \times (r + 1)$ matrix (each column is an evaluation point). $\boldsymbol{w}$ gives weights.

1: **function** $(\boldsymbol{R}, \boldsymbol{w}) = \mathbf{simplex}(r)$
2:      $\alpha = \sqrt{r + 1}$
3:      $\beta = (1 + \alpha)^{-1}$
4:      $\boldsymbol{R} = [\alpha \boldsymbol{I}_{r \times r} - [\beta]_{r \times r} \quad [-1]_{r \times 1}]$
5:      $\boldsymbol{w} = \left[\frac{1}{r+1}\right]_{r \times 1}$
6: **end function**

---

**Algorithm 2** Cross-polytope vertices, symmetric reference quadrature

---

**Input:** $r \in \mathbb{Z}_{\geq 1}$ is the number of reference dimensions.
**Output:** $\boldsymbol{R}$ is an $r \times 2r$ matrix (each column is an evaluation point). $\boldsymbol{w}$ gives weights.

1: **function** $(\boldsymbol{R}, \boldsymbol{w}) = \mathbf{cross}(r)$
2:      $\boldsymbol{R} = \sqrt{r} [\boldsymbol{I}_{r \times r} \quad -\boldsymbol{I}_{r \times r}]$
3:      $\boldsymbol{w} = \left[\frac{1}{2r}\right]_{2r}$
4: **end function**

---

**Algorithm 3** Random-affinity sigma points

---

**Input:** $d$ is the number of parameter dimensions and $r$ is the number of reference dimensions.
**Output:** $\boldsymbol{X}$ is a $d \times n$ matrix (each column contains an evaluation point). $\boldsymbol{w}$ gives weights so that

$$R[f] = \sum_{k=1}^{n} \boldsymbol{w}_k f\left(\boldsymbol{x}^{(k)}\right) \approx \int f(\boldsymbol{x}) \, d\mathcal{N}(\boldsymbol{x} \mid 0, \boldsymbol{I}). \tag{19}$$

1: **function** $(\boldsymbol{X}, \boldsymbol{w}) = \mathbf{rasp}(d, r)$
2:      $(\boldsymbol{R}, \boldsymbol{w}) = \mathbf{ref\_quad}(r)$                    ▷ Get reference quadrature.
3:      $\boldsymbol{S}^T \leftarrow \begin{bmatrix} \boldsymbol{R}^T & -\boldsymbol{R}^T \end{bmatrix}^T$         ▷ Concatenate with negative coordinates.
4:      $\boldsymbol{j} = \lfloor 2r \, \mathbf{rand}_{d \times 1} \rfloor + 1$       ▷ Randomize mapping to signed reference dimensions.
5:      $\boldsymbol{X}_{ik} = \boldsymbol{S}_{j_i k}$      for $i \in [d]$ and $k \in [n]$.         ▷ Compose each evaluation point $k$.
6: **end function**

---

## A.2  EXACTNESS PROOFS AND VARIANCE COMPARISONS

**Proof of Theorem 3.1**    Consider a function that is polynomial in a single parameter, i.e $f(\boldsymbol{\theta}) \equiv p(\boldsymbol{\theta}_i)$, of degree less than or equal to $\varepsilon_1$. From Equation (13), the RASP quadrature evaluates to

$$Q[f] = \sum_{k=1}^{n} w_k f(\boldsymbol{\mu} + \boldsymbol{\sigma} * \boldsymbol{x}^{(k)}) = \sum_{k=1}^{n} w_k p(\boldsymbol{\mu}_i + \boldsymbol{\sigma}_i \boldsymbol{x}_i^{(k)}) = \sum_{k=1}^{n} w_k p(\boldsymbol{\mu}_i + \boldsymbol{\sigma}_i s_i \boldsymbol{\rho}_{j_i}^{(k)}).$$

Since $R[\cdot]$ has univariate exactness $\varepsilon_1$, applying Equations (12) and (14) gives

$$Q[f] = \int p(\boldsymbol{\mu}_i + \boldsymbol{\sigma}_i s_i \boldsymbol{\rho}_{j_i}) d\mathcal{N}(\boldsymbol{\rho} \mid 0, \boldsymbol{I}).$$

The affine transformation $x = \boldsymbol{\mu}_i + \boldsymbol{\sigma}_i s_i \boldsymbol{\rho}_{j_i}$ yields the matching density $\mathcal{N}(x \mid \boldsymbol{\mu}_i, \boldsymbol{\sigma}_i^2)$. Thus

$$Q[f] = \int p(x) d\mathcal{N}(x \mid \boldsymbol{\mu}_i, \boldsymbol{\sigma}_i^2) = \int f(\boldsymbol{\theta}) d\mathbf{q}(\boldsymbol{\theta} \mid \boldsymbol{\mu}, \boldsymbol{\sigma}^2) \quad \square$$

**Proof of Theorem 3.2** Consider a function that is polynomial in the two parameter dimensions, i.e. $f(\boldsymbol{\theta}) \equiv p(\boldsymbol{\theta}_{i_1}, \boldsymbol{\theta}_{i_2})$, of total degree less than or equal to $\varepsilon_2$. From Equation (13), the RASP quadrature evaluates to

$$Q[f] = \sum_{k=1}^{n} w_k f(\boldsymbol{\mu} + \boldsymbol{\sigma} * \boldsymbol{x}^{(k)}) = \sum_{k=1}^{n} w_k p(\boldsymbol{\mu}_{i_1} + \boldsymbol{\sigma}_{i_1} \boldsymbol{x}_{i_1}^{(k)}, \boldsymbol{\mu}_{i_2} + \boldsymbol{\sigma}_{i_2} \boldsymbol{x}_{i_2}^{(k)})$$

$$= \sum_{k=1}^{n} w_k p(\boldsymbol{\mu}_{i_1} + \boldsymbol{\sigma}_{i_1} s_{i_1} \boldsymbol{\rho}_{j_{i_1}}^{(k)}, \boldsymbol{\mu}_{i_2} + \boldsymbol{\sigma}_{i_2} s_{i_2} \boldsymbol{\rho}_{j_{i_2}}^{(k)})$$

The reference quadrature will be exact if the corresponding reference dimensions are distinct. Taking $j_1 = j_{i_1}$ and $j_2 = j_{i_2}$, we have $\mathbf{p}(j_1 = j_2) = \frac{1}{r}$, since they are assigned uniformly at random over $r$ outcomes. Thus, the probability that we obtain the same exactness is $\frac{r-1}{r}$. In this case, the polynomial remains bivariate in the reference space. Since total degree does not change, we can apply the bivariate exactness $\varepsilon_2$ so that Equation (14) gives

$$Q[f] = \int p(\boldsymbol{\mu}_{i_1} + \boldsymbol{\sigma}_{i_1} s_{i_1} \boldsymbol{\rho}_{j_1}, \boldsymbol{\mu}_{i_2} + \boldsymbol{\sigma}_{i_2} s_{i_2} \boldsymbol{\rho}_{j_2}) d\mathcal{N}(\boldsymbol{\rho} \mid 0, \boldsymbol{I}).$$

As before, the affine transformations $x_1 = \boldsymbol{\mu}_{i_1} + \boldsymbol{\sigma}_{i_1} s_{i_1} \boldsymbol{\rho}_{j_1}$ and $x_2 = \boldsymbol{\mu}_{i_2} + \boldsymbol{\sigma}_{i_2} s_{i_2} \boldsymbol{\rho}_{j_2}$ give densities $\mathcal{N}(x_1 \mid \boldsymbol{\mu}_{i_1}, \boldsymbol{\sigma}_{i_1}^2)$ and $\mathcal{N}(x_2 \mid \boldsymbol{\mu}_{i_2}, \boldsymbol{\sigma}_{i_2}^2)$. Thus

$$Q[f] = \int p(x_1, x_2) \, d\mathcal{N}(x_1 \mid \boldsymbol{\mu}_{i_1}, \boldsymbol{\sigma}_{i_1}^2) \, d\mathcal{N}(x_2 \mid \boldsymbol{\mu}_{i_2}, \boldsymbol{\sigma}_{i_2}^2) = \int f(\boldsymbol{\theta}) d\mathbf{q}(\boldsymbol{\theta} \mid \boldsymbol{\mu}, \boldsymbol{\sigma}^2) \,\square$$

**Proof of Theorem 3.3** The bivariate polynomial terms that are not correctly integrated by $Q[\cdot]$ occur when both parameters map to the same reference dimension, i.e. when both $i_1$ and $i_2$ are assigned to the same index, $j_{i_1} = j_{i_2} = j$. In this case, Equation (13) gives

$$Q[f] = \sum_{k=1}^{n} w_k f(\boldsymbol{\mu} + \boldsymbol{\sigma} * \boldsymbol{x}^{(k)}) = \sum_{k=1}^{n} w_k (\boldsymbol{\sigma}_{i_1} \boldsymbol{x}_{i_1}^{(k)})(\boldsymbol{\sigma}_{i_2} \boldsymbol{x}_{i_2}^{(k)})$$

$$= \sum_{k=1}^{n} w_k s_{i_1} s_{i_2} \boldsymbol{\sigma}_{i_1} \boldsymbol{\sigma}_{i_2} \boldsymbol{\rho}_j^{(k)2} = s_{i_1} s_{i_2} \boldsymbol{\sigma}_{i_1} \boldsymbol{\sigma}_{i_2}.$$

Since $\mathbb{E}[s_{i_1} s_{i_2}] = 0$, it follows $\mathbb{E}\left[Q[f]\right] = \int f(\boldsymbol{\theta}) d\mathbf{q}(\boldsymbol{\theta} \mid \boldsymbol{\mu}, \boldsymbol{\sigma}^2) = 0 \,\square$

**Proof of Theorem 3.4** From Equation (13), the RASP quadrature evaluates to

$$Q[f] = \sum_{k=1}^{n} w_k f(\boldsymbol{\mu} + \boldsymbol{\sigma} * \boldsymbol{x}^{(k)}) = \sum_{k=1}^{n} w_k \prod_{i=1}^{d} \left(\boldsymbol{\sigma}_i \boldsymbol{x}_i^{(k)}\right)^{m_i} = \sum_{k=1}^{n} w_k \prod_{i=1}^{d} \left(\boldsymbol{\sigma}_i s_i \boldsymbol{\rho}_{j_i}^{(k)}\right)^{m_i}$$

We can sum each term twice and halve the result. Then, applying symmetry gives

$$Q[f] = \frac{1}{2} \left[ \sum_{k=1}^{n} w_k \prod_{i=1}^{d} \left(\boldsymbol{\sigma}_i s_i \boldsymbol{\rho}_{j_i}^{(k)}\right)^{m_i} + \sum_{k'=1}^{n} w_{k'} \prod_{i=1}^{d} \left(\boldsymbol{\sigma}_i s_i \boldsymbol{\rho}_{j_i}^{(k')}\right)^{m_i} \right]$$

$$= \frac{1}{2} \left[ \sum_{k=1}^{n} w_k \prod_{i=1}^{d} \left(\boldsymbol{\sigma}_i s_i \boldsymbol{\rho}_{j_i}^{(k)}\right)^{m_i} + \sum_{k=1}^{n} w_k \prod_{i=1}^{d} (-1)^{m_i} \left(\boldsymbol{\sigma}_i s_i \boldsymbol{\rho}_{j_i}^{(k)}\right)^{m_i} \right].$$

Since $\sum_{i=1}^{d} m_i$ is odd, we have $\prod_{i=1}^{d} (-1)^{m_i} = -1$ and every term cancels, thus recovering

$$Q[f] = \int \prod_{i=1}^{d} (\boldsymbol{\theta}_i - \boldsymbol{\mu}_i)^{m_i} d\mathbf{q}(\boldsymbol{\theta} \mid \boldsymbol{\mu}, \boldsymbol{\sigma}^2) = 0 \,\square \tag{20}$$

| Error Variance $\parallel$ | $\frac{\boldsymbol{\theta}_i - \boldsymbol{\mu}_i}{\boldsymbol{\sigma}_i}$ | $\frac{(\boldsymbol{\theta}_i - \boldsymbol{\mu}_i)^2}{\boldsymbol{\sigma}_i^2}$ | $\frac{(\boldsymbol{\theta}_i - \boldsymbol{\mu}_i)(\boldsymbol{\theta}_j - \boldsymbol{\mu}_j)}{\boldsymbol{\sigma}_i \boldsymbol{\sigma}_j}$ |
|---|---|---|---|
| Monte Carlo $\parallel$ | $\frac{1}{n}$ | $\frac{2}{n}$ | $\frac{1}{n}$ |
| RASP $\parallel$ | $0$ | $0$ | $\frac{1}{r}$ |

Table 1: Variance of errors for quadratic basis functions. RASP quadratures are more efficient for univariate basis functions, but have higher variance than Monte Carlo for bivariate basis functions.

### A.3 TRAINING METHODS COMPARISONS

Figures 8 to 10 compare the optimal validation quadratures for SGVB and QNVB against standard training algorithms, including: stochastic gradient descent with momentum (SGD-M) (Qian, 1999), Adam (Kingma & Ba, 2014), AdaHessian (Yao et al., 2021), and AdaGrad (Duchi et al., 2011). These experiments show that QNVB using RASP quadratures provides a competitive training algorithm.

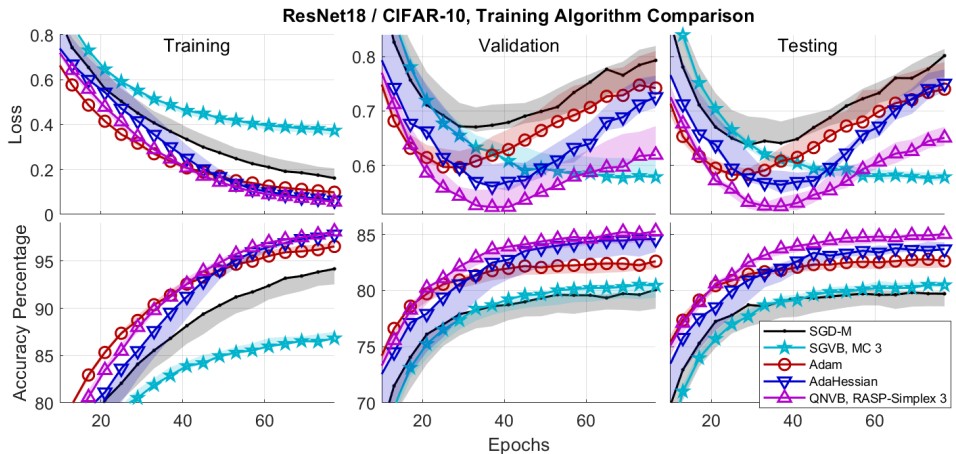

Figure 8: We compare QNVB with RASP-simplex against other training algorithms. QNVB with RASP-simplex integration achieves better generalizability on the test data.

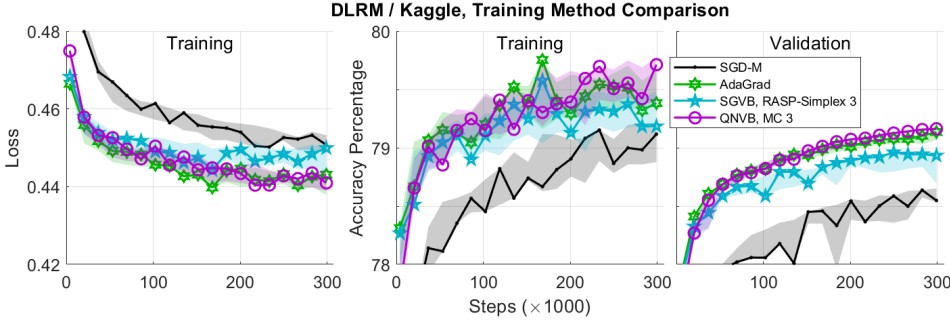

Figure 9: We examine the performance of QNVB with Monte Carlo integration in comparison to alternative training methods. QNVB still achieves better generalizability on validation data.

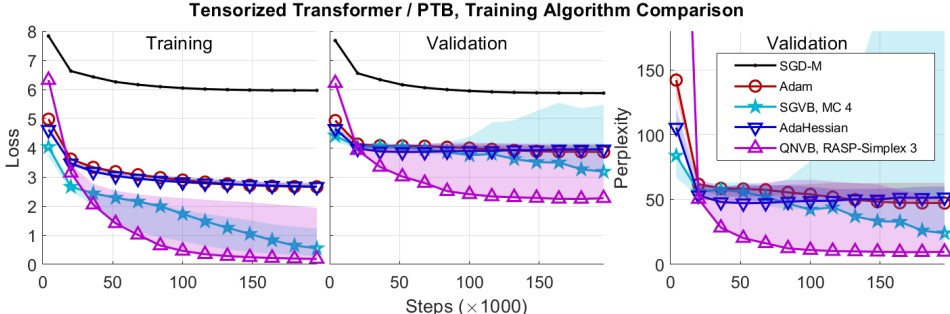

Figure 10: We present a comparison of training algorithms on the Tensorized Transformer. QNVB with RASP-simplex achieves much lower perplexity than the alternatives.

## A.4 EXPERIMENT DETAILS

The hyperparameter settings for QNVB and SGVB follow those used by Duersch (2023), for all experiments. Other settings for SGD-M (Qian, 1999), Adam (Kingma & Ba, 2014), AdaHessian (Yao et al., 2021), and AdaGrad (Duchi et al., 2011) follow the setup used by Yao et al. (2021), except we extended the training duration for the Tensorized Transformer to $200,000$ steps. These experiments do not use regularization, which is equivalent to an improper flat prior.

**ResNet18 / CIFAR-10**   The version of ResNet18 we tested is from PyTorch TorchVision 0.15.1 library. Training data were split into $40,000$ training cases and $10,000$ validation cases. SGD-M uses the learning rate $\lambda = 0.1$ and gradient momentum coefficient of $\beta_1 = 0.9$; Adam uses the standard learning rate $\lambda = 10^{-3}$ and betas, $\beta_1 = 0.9$ and $\beta_2 = 0.999$, and the denominator coefficient $\varepsilon = 10^{-8}$. AdaHessian uses the same hyperparameters as Adam, except with the standard learning rate for AdaHessian of $\lambda = 0.15$. Both SGVB and QNVB use the same learning schedule from Duersch (2023). Namely, QNVB uses $\lambda = 5 \times 10^{-3}$, the same beta and epsilon parameters as Adam, and a learning rate reduction schedule of $1.05$ per epoch. SGVB uses the same hyperparameters as Adam and the same standard deviation limits as QNVB. QNVB uses the standard deviation limits $\sigma_{\min} = 10^{-3}$ and $\sigma_{\max} = 5 \times 10^{-2}$, and the likelihood weight $w = 40,000$. Each training run consisted of 80 epochs.

**DLRM / Criteo Ad Kaggle**   For these experiments, SGD-M, AdaGrad, SGVB, and QNVB were compared. AdaHessian was not tested because this architecture relies on sparse gradients during training, which are not compatible with secondary backpropagation, and the requisite modifications used by Yao et al. (2021) were not available. Both SGD-M and AdaGrad use standard DLRM hyperparameters. SGVB uses standard Adam hyperparameters, as well as $\sigma_{\min} = 10^{-5}$ and $\sigma_{\max} = 10^{-3}$. QNVB uses $\lambda = 2 \times 10^{-3}$, $w = 5 \times 10^{-4}$, and likelihood annealing. In particular, the likelihood weight is increased by a factor of $1.0000248$ per step to arrive at a final weight of $w = 10^8$. Both SGD-M and AdaGrad require far less training time due to their usage of sparse gradients to avoid updating all parameters in each step. Training consists of 1 epoch, approximately 300,000 steps, as this architecture immediately exhibits over training at the start of the second epoch.

**Tensorized Transformer / PTB**   The learning rates are $\lambda = 5 \times 10^{-4}$ (SGD-M), $\lambda = 2.5 \times 10^{-4}$ (Adam), and $\lambda = 1$ (AdaHessian). SGD-M has the gradient moment coefficient set to 0.9, while both SGD-M and Adam use a cosine learning rate schedule. Training consists of 200,000 steps.

