# OpenReview forum: "RASP Quadratures: Efficient Numerical Integration for High-Dimensional Mean-Field Variational Inference"
_ICLR.cc/2024/Conference — Submitted to ICLR 2024_

### Official Review · Reviewer_m4a3 · 2023-10-30

**Soundness:** 2 fair
**Presentation:** 2 fair
**Contribution:** 2 fair
**Rating:** 6
**Confidence:** 2

**Summary:**

The authors proposed RASP quadratures which serve as an evaluation-efficient method for high-dimensional integration. RASP quadratures minimize the error corresponding to basis functions that dominate variational updates for Gaussian mean fields.

**Strengths:**

The idea of  using RASP quadrature for QNVB is interesting and new.

The performance of the proposed method is demonstrated through a range of experiments.

**Weaknesses:**

The related work section should be expanded to include more discussion. Clearly, QMC is very relevant and other quadrature methods are also relevant.

**Questions:**

Is there any theoretical justification for the number of nodes (and corresponding function evaluations) in general?

---

> ### Author Response · Authors · 2023-11-17
>
> Thank you for taking the time to review our work. We really appreciate your positive feedback.
>
> ### “The related work section should be expanded to include more discussion. Clearly, QMC is very relevant and other quadrature methods are also relevant.”
>  - Agreed. We substantially revised the related work section to provide a more thorough overview these methods to help place our work in a clearer context.
>
> ### “Is there any theoretical justification for the number of nodes (and corresponding function evaluations) in general?”
>  - Theorem 2 shows how the number of nodes affects the number of bivariate quadratic basis functions that integrate exactly. That said, the optimal number of nodes really depends on time / computational constraints and the characteristics of the loss topography.

---

> > ### Comment · Reviewer_m4a3 · 2023-11-22
> >
> > I have read the authors' response and decide to keep my score.

---

### Official Review · Reviewer_jXiK · 2023-10-31

**Soundness:** 1 poor
**Presentation:** 2 fair
**Contribution:** 2 fair
**Rating:** 3
**Confidence:** 4

**Summary:**

This paper proposes a new randomized quadrature scheme, random-affinity sigma-point (RASP) quadrature, for the purpose of mean-field variational inference (MFVI). RASP is combined with a recent variational inference method proposed in a separate paper based on a method called projective integral updates. The resulting method essentially combines RASP with MFVI using quasi-newton-like preconditioning.

**Strengths:**

* The paper proposes a new randomized quadrature scheme they call RASP.

**Weaknesses:**

The paper lacks reference to a number of highly relevant related work that develops a context for the current paper. Here are some places (non-exhaustive list) where I think a citation is needed, with specific recommendations:
* Section 1 first paragraph: cite something for Markov-chain Monte Carlo (MCMC). Although this is folk wisdom, a citation is required for the claim "MCMC is not practical for large learning architectures that often contain billions of parameters."
* Section 1 second paragraph: cite something for mean-field variational inference (MFVI). MFVI didn't appear out of thin air. The development/popularization of MFVI is often attributed to [1-3].
* Section 1.2 last sentence: stochastic gradient variational Bayes (SGVB) was independently developed by [4-6], where the use of SGVB for general Bayesian inference is commonly attributed to [4,5]. I suspect the intended citation here was Kingma and Welling [6]?
* Section 2.1 first sentence: cite something for variational inference. In general, people cite these two well-known reviews [7,8].

Probably due to the issue above, the paper misses a very important line of related works:
* Randomized quasi-Monte Carlo (RMQC) for variational inference [9]
* RQMC and stochastic quasi-Newton optimization for variational inference [10]

While I'm not claiming that the method lacks novelty, this paper does not do a good job of putting the work in the context of this line of work. Does this method improve over these works? Is it different from the (randomized) quasi-Monte Carlo schemes that are used in these works? This leads me to the next point.

The choice of baselines and the overall design of the experiments makes the conclusion of the experiments unclear. In particular, the experiments do not appear to be doing an apple-to-apple comparison. Also, some experimental details are missing.
* Variational inference is fundamentally a method for approximating the posterior. How was the joint likelihood obtained here? Was a prior set on the parameters of these models? Was an improper flat prior set on the parameters? What were the hyperparameters? It is, in fact, known that the choice of prior in deep learning models affects the result a lot [11].
* The paper only compares against the following combinations: RASP + QNVB, MC + SGVB, MC + QNVB, QMC + QNVB. Why was RASP + SGVB not compared against? Also, the closely related works based on randomized QMC [9,10] and the QNVB algorithm proposed in [10] should also have been included.


### Minor Comments
* When denoting the variational approximation q, it is non-standard to denote conditioning on $\varphi$. This is because the variational parameters $\varphi$ are not considered to be random variables (their probability density has not been defined)
* Section 1.1 third paragraph second from the last sentence “... Hessian diagonal of the training loss, which can be expensive to compute.” I believe saying that it is noisy is more accurate. In fact, noisy estimates are cheap to obtain, as shown by [12].
* Section 2.2 last sentence: is there any evidence that alternative integration schemes scale better in terms of dimensionality? Because quasi-Monte Carlo methods, while converging faster with respect to the number of samples, have an explicit dimension dependence, unlike regular Monte Carlo. I suspect RASP would have similar behavior, although it is hard to tell since the author did not present a convergence bound akin to those in the QMC literature. Nonetheless, when the number of samples is small, I am not sure if the method can be claimed to be scalable in terms of dimension.

### References
I am not the author of nor affiliated with any of the references below:
1. Anderson, James R., and Carsten Peterson. "A mean field theory learning algorithm for neural networks." Complex Systems 1 (1987): 995-1019.
2. Peterson, Carsten, and Eric Hartman. "Explorations of the mean field theory learning algorithm." Neural Networks 2.6 (1989): 475-494.
3. Hinton, Geoffrey E., and Drew Van Camp. "Keeping the neural networks simple by minimizing the description length of the weights." Proceedings of the sixth annual conference on Computational learning theory. 1993.
4. Ranganath, Rajesh, Sean Gerrish, and David Blei. "Black box variational inference." Artificial intelligence and statistics. PMLR, 2014.
5. Titsias, Michalis, and Miguel Lázaro-Gredilla. "Doubly stochastic variational Bayes for non-conjugate inference." International conference on machine learning. PMLR, 2014.
6. Kingma, Diederik P., and Max Welling. "Auto-encoding variational bayes." ICLR'14.
7. Jordan, Michael I., et al. "An introduction to variational methods for graphical models." Machine learning 37 (1999): 183-233.
8. Blei, David M., Alp Kucukelbir, and Jon D. McAuliffe. "Variational inference: A review for statisticians." Journal of the American statistical Association 112.518 (2017): 859-877.
9. Buchholz, Alexander, Florian Wenzel, and Stephan Mandt. "Quasi-monte carlo variational inference." International Conference on Machine Learning. PMLR, 2018.
10. Liu, Sifan, and Art B. Owen. "Quasi-monte carlo quasi-newton in variational bayes." The Journal of Machine Learning Research 22.1 (2021): 11043-11065.
11. Fortuin, Vincent, et al. "Bayesian neural network priors revisited." arXiv preprint arXiv:2102.06571 (2021).
12. Fan, Kai, et al. "Fast second order stochastic backpropagation for variational inference." Advances in Neural Information Processing Systems 28 (2015).

**Questions:**

no questions

---

> ### Author Response · Authors · 2023-11-17
>
> Thank you for taking the time to review this work. We genuinely appreciate the obvious effort and expertise you put into your review. We have made a diligent effort to revise the manuscript to be as responsive as possible. We inappropriately discounted the importance of maintaining needed citations to place our work in context and have made an earnest effort to correct this.
>
> ### “The paper lacks references to a number of highly relevant related work... [Here are] specific recommendations:”
>  - Thank you for these citations. We carefully reviewed this work and performed additional searching to provide other relevant sources as well.
>
> ### “Cite something for the claim ‘MCMC is not practical for large learning architectures.’"
>  - We added several references to support this claim.
>
> ### “Cite something for mean-field variational inference (MFVI)... often attributed to [1-3].”
>  - We were aware of these references and should have added them earlier.
>
> ### “SGVB was independently developed by [4-6]... I suspect the intended citation here was Kingma and Welling [6]?
>  - Yes, we corrected this oversight [6] and added [4,5].
>
> ### “Cite something for VI... [7,8].”
>  - Thank you. We added these well-known sources and a recent survey by Zhang (2018).
>
> ### “The paper misses a very important line of related works [regarding RQMC].”
>  - We appreciate these references and we also performed additional searching to discuss these contributions in more detail under related work.
>
> ### “While I'm not claiming that the method lacks novelty, this paper does not do a good job of putting the work in context.”
>  - The related work section has been heavily revised to provide a clearer narrative and context.
>
> ### “The choice of baselines ... makes the conclusion of the experiments unclear... the experiments do not appear to be doing an apple-to-apple comparison. Why was RASP + SGVB not compared? Also, some details are missing.”
>  - RASP quadratures target improved integration of basis functions that dominate QNVB updates. This is why our original experiments examined several architectures and other variance-reduced monte carlo methods with QNVB. That said, a quadrature comparison with other training methods is helpful and we added a battery of tests using SGVB, as suggested. We had to move comparisons of non-VI training methods into the appendix, but that isn't problematic since it isn’t the central focus of the paper. We also added additional experiment details to the appendix.
>
> ### “Was a prior set on the parameters of these models? What were the hyperparameters?"
>  - We used uniform priors to avoid an additional complication, but regularization techniques can easily be incorporated into our pipeline. As you are almost certainly aware, L1 and L2 terms can be regarded as negative log priors. Other hyperparameters are described in the appendix. We added a paragraph about this under the experiments section.
>
> ### “Also, the closely related works based on randomized QMC ... should also have been included."
>  - RQMC methods typically use more function evaluations than our target range (10+ as opposed to 3 to 6) and they are not designed to target second-order exactness. Such comparisons would be interesting, but they also open a different line of investigation from the role of exactness in QNVB updates. Please see our updated related work.
>
> ### “When denoting the variational approximation q, it is non-standard to denote conditioning on phi.”
>  - Our notation follows Zhang (2018) to explicitly clarify that the shape of the variational density is determined by phi, which is the actual variable of optimization. This notation is helpful for projective integral updates, since phi comprises the projection coefficients that RASP quadratures target. See equations 3 and 4.
>
> ### “Section 1.1 ‘... Hessian diagonal of the training loss, which can be expensive to compute.’ I believe saying that it is noisy is more accurate. In fact, noisy estimates are cheap to obtain, as shown by [12].”
>  - This is a fair point and we have made the modification.
>
> ### “Section 2.2: is there any evidence that alternative integration schemes scale better in terms of dimensionality? Because quasi-Monte Carlo methods ... have an explicit dimension dependence. I suspect RASP would have similar behavior... the author did not present a convergence bound... I am not sure if the method can be claimed to be scalable in terms of dimension.”
>  - RASP quadratures are not unbiased estimators and they do not converge to the exact integral with increasing points. While they are exact on all linear basis functions, as well as a large fraction of second-order basis functions (theorems 1 and 2), there are systematic errors that persist for some higher-order basis functions. The point of interest regarding increased reference dimensionality is the fraction of second-order basis functions that become exact, which we have analyzed with theorems and proofs.

---

> ### Comment · Reviewer_jXiK · 2023-11-18
> **Response**
>
> I thank the authors for their response and the updated manuscript. Here, I will highlight some points that I am in disagreement with the authors. My primary concern is that, due to the rather unusual setup of the theory and experiments, it is very hard to judge the benefit of the proposed methods compared to existing solutions in the literature.
>
> ## On experimental setup
> Firstly, I believe the authors would argue that the main use case of variational inference is Bayesian inference. Unfortunately, none of the presented experiments can be considered a "typical" Bayesian inference task. The problem is that, for a typical VI person like me, the experimental setup does not resonate with the typical setup of VI. Sure, Bayesian deep learning is a thing, but it is a rather specialized setting where VI has, time and time again, appeared to not be very competitive compared to other methods. There is also theoretical evidence that mean-field VI may fundamentally be incompatible with deep learning [1]. Thus, I recommend the authors take a look at any of the influential variational inference methodology papers to see what are the typical benchmarks and metrics used for evaluation. (Typically, we compare evidence lower bounds. I believe what is referred to as the "loss" here is something different?)
>
> ## On the theoretical results
> Currently, the presentation of the theoretical results is unconventional, and it is also hard to compare against existing convergence/error guarantees for MC and QMC methods. Note that it is also the task of the authors to highlight and compare the theoretical guarantees against previous guarantees if they wish to claim theoretical superiority.
>
> ## Comparison Against QMCVI, QNQMCVI
> > RQMC methods typically use more function evaluations than our target range (10+ as opposed to 3 to 6) and they are not designed to target second-order exactness. Such comparisons would be interesting, but they also open a different line of investigation from the role of exactness in QNVB updates. Please see our updated related work.
>
> I do not quite follow the response here. QMCVI and QNQMCVI are separate algorithms on their own. I'm not asking to combine RQMC with "QNVB" by Duerch (2023). I am suggesting to compare against the exact algorithm proposed by Liu and Owen (2021) and Buchholz et al. (2018). Especially, if the authors claim that the proposed method performs better with less computation than QMCVI and QNQMCVI, then they should present such evidence!
>
> ## References
> 1. Coker, Beau, Bruinsma, Wessel P., Burt, David R., Pan, Weiwei, & Doshi-Velez, Finale. 2022. Wide Mean-Field Bayesian Neural Networks Ignore the Data. AISTATS.

---

> > ### Author Response · Authors · 2023-11-21
> >
> > We appreciate your perspective and critical engagement and we have made a diligent effort to respond to your concerns in good faith. Perhaps a key point of disagreement centers on the scope of the paper. As you know, the length limitation on ICLR papers is necessary to keep exposition concise and content focused. In this case, the principal question is whether a quadrature that integrates a large fraction of second-order basis functions exactly can improve high-dimensional VI for large models, when every function and gradient evaluation can be quite costly. We have proposed a simple method to create a new class of high-dimensional quadratures that limits memory usage. We have also rigorously characterized the exactness properties, which directly addresses the line of reasoning that led to this quadrature formulation. The purpose of this paper is not to compare QNVB to other VI or quasi-Newton methods (including QMCVI and QNQMCVI); it is to compare integration methods that use very few evaluations. Since QNVB steps directly incorporate projection onto univariate second-order basis functions, which RASP exactness targets, we had good reason to anticipate RASP quadratures would have a non-trivial effect on QNVB optimization and our experiments bear this out.
> >
> > The additional point regarding [1] is interesting, but, in our view, this new objection is out of scope. Their primary analysis applies to fully connected networks with odd activation functions, standard normal priors, and in the limit that the width tends to infinity. To suggest that MFVI is fundamentally incompatible with deep learning is a misrepresentation of their position, which is more nuanced and only advises caution. Further, our experiments (now in the appendix to accommodate additional SGVB + RASP comparisons) show that MFVI can be competitive with standard deep learning training algorithms.
> >
> > Regarding convergence analysis, the difficulty is that RASP quadratures pursue a qualitatively distinct objective from QMC: exactness versus convergence. The exactness of QMC methods is zeroth order, i.e. the only basis function that integrates exactly is 1. In contrast, RASP quadratures are all exact to second order in univariate basis functions, with an additional fraction of bivariate basis functions that depends on the precise method. Since RASP quadratures do not (nor are intended to) converge to the exact integral, convergence analysis is not possible. Ultimately, the metric that is arguably more important than convergence is prediction quality on unseen data when time and computation must adhere to severe limits, which our experiments measure.

---

### Official Review · Reviewer_rgHx · 2023-11-10

**Soundness:** 4 excellent
**Presentation:** 3 good
**Contribution:** 2 fair
**Rating:** 6
**Confidence:** 3

**Summary:**

This paper proposes a "random-affinity sigma-point (RASP) quadrature" method for numerical integration w.r.t. high-dimensional Gaussian distributions. The point is to use very few integration nodes. RASP quadrature is constructed by "scaling" a quadrature method on a low-dimensional reference space to higher dimensions by retaining the quadrature weights and picking random coordinates (and doing sign-changes) of the original quadrature nodes.

RASP quadrature is applied to some large-scale problems that I am not qualified to comment on. In this review I focus on the quadrature part of the paper.

**Strengths:**

The paper is well and clearly written. In some of the experiments the proposed method is observed to outperform some competitors. I suppose the RASP quadrature being quite simply can also be seen as a strength.

**Weaknesses:**

I am rather skeptical about novelty of the proposed quadrature. Having never worked in a setting in which even n = d + 1 nodes is prohibitive, I cannot provide any references. However, the limited and superficial literature review on pages 4 and 9 does not convince me that the authors know the relevant quadrature well enough to say with any confidence that something like this has not been tried before.

I would like to see some language reflecting this inserted in the paper.

However, the quadrature method does not necessarily need to be novel. I am not against accepting this paper if the consensus is that the overall variational inference methodology presented here constitutes a useful contribution to the field.

**Questions:**

- p3: What is [m]?
- p3: The definition of \mathcal{F} in Eq (3) is unclear.
- p3: There are several squares of bolded quantities, some of which appear to be vectors [e.g., in Eq (5)]. What does squaring mean here?
- p3: In the first paragraph of Sec 2.3 f: R^d \mapsto \R should use \to rather than \mapsto.
- p4: "For quadrature formulas that obtain second-order exactness (integrating all quadratic total-degree polynomials exactly) over multivariate Gaussians, the evaluation nodes are called sigma points." It would be good to note that sigma point terminology is far from universal and only really used in non-linear filtering literature.
- p4: Given McNamee & Stenger (1967) and other old fully symmetric quadrature formulae, I doubt that Uhlmann (1995) was the first to design a second-order accurate quadrature rule.
- p9: The description of QMC in Section 5 is rather misleading (as is labelling some alternative methods as QMC methods in Figure 4): QMC methods share nothing common with MC methods beyond the fact that they use uniform integration weights w_k = 1/n. See e.g. p135 in Dick et al. (2013).

**Details Of Ethics Concerns:**

-

---

> ### Author Response · Authors · 2023-11-17
>
> Thank you for taking the time to review our work. We really appreciate your effort and expertise.
>
> ### “The limited and superficial literature review on pages 4 and 9 does not convince me that the authors know the relevant quadrature well enough to say with any confidence that something like this has not been tried before.”
>  - This is a fair point, and we inappropriately disregarded the importance of citing relevant related work. We have diligently reviewed surveys, books, and new papers regarding high-dimensional integration, sigma-point methods, QMC methods, and RQMC methods. We have also substantially revised and lengthened our related work section accordingly to coherently place our contributions in context. While we still cannot be sure if this method (or something similar) has been published before, it is certainly not commonly discussed. We have added a statement to this effect.
>
> ### “What is [m]?”
>  - This notational shorthand is sometimes used to denote the set {1, 2, … , m}.
>
> ### “What does squaring [of vectors] mean here?”
>  - Powers on vectors are elementwise in these formulas and we have added a note to clarify this notation.
>
> ### “The definition of \mathcal{F} in Eq (3) is unclear.”
>  - The basis depends on the exponential family that captures feasible variational densities. For the purpose of QNVB, the basis spans the set of univariate quadratics in R^d, which Eq (5) is intended to capture.
>
> ### “In the first paragraph of Sec 2.3 f: R^d \mapsto \R should use \to rather than \mapsto.”
>  - Thank you!
>
> ### "It would be good to note that sigma point terminology is far from universal and only really used in non-linear filtering literature.”
>  - Good point. We have added this clarification.
>
> ### “Given McNamee & Stenger (1967) and other old fully symmetric quadrature formulae, I doubt that Uhlmann (1995) was the first to design a second-order accurate quadrature rule.”
>  - Another good point; we have added this citation and removed the “original” qualifier.
>
> ### “The description of QMC in Section 5 is rather misleading (as is labelling some alternative methods as QMC methods in Figure 4): QMC methods share nothing common with MC methods beyond the fact that they use uniform integration weights w_k = 1/n. See e.g. p135 in Dick et al. (2013).”
>  - We also reviewed several sources and realize that it was a mistake to label variance-reduced methods as QMC methods. Our confusion came from misreading Caflisch’s 1998 paper, “Monte carlo and quasi-monte carlo methods.” While his paper is clear about the definition of QMC, it also contains variance-reduce methods, which are neither MC nor QMC. We have corrected the language and experiments to use the more explicit and technically correct terminology of variance-reduced Monte Carlo (VRMC).

---

### Meta-Review · Area_Chair_Z9oT · 2023-12-06

**Metareview:**

This paper provides a novel algorithm for quadrature in a very specific setting (extremely high dimensionality / high cost of function evaluations).

The reviewers generally agreed that the idea and algorithm are novel and indeed conceptually unusual. They also all raised concerns about the limited literature review / contextualization, and the limited motivation for the rather special domain setup. After the discussion between authors and reviewers -- which, I want to note, was exemplarily civil and reflective -- the reviewers agreed in their internal discussion that the paper, while valuable, needs further work to be acceptable. I can thus not recommend it for acceptance in the current form.

**Justification For Why Not Higher Score:**

The paper is correct, but just rather niche. There is also a realistic concern that the authors may have missed related approaches from other communities, which was flagged by two reviewers.

**Justification For Why Not Lower Score:**

N/A

---

### Decision · Program_Chairs · 2024-01-16

Reject